# Zebrafish preserve global germline DNA methylation while sex-linked rDNA is amplified and demethylated during feminisation

Oscar Ortega-Recalde [1], Robert C. Day[2], Neil J. Gemmell [1] & Timothy A. Hore [1]

The germline is the only cellular lineage capable of transferring genetic information from one generation to the next. Intergenerational transmission of epigenetic memory through the germline, in the form of DNA methylation, has been proposed; however, in mammals this is largely prevented by extensive epigenetic erasure during germline definition. Here we report that, unlike mammals, the continuously-defined 'preformed' germline of zebrafish does not undergo genome-wide erasure of DNA methylation during development. Our analysis also uncovers oocyte-specific germline amplification and demethylation of an 11.5-kb repeat region encoding 45S ribosomal RNA (fem-rDNA). The peak of fem-rDNA amplification coincides with the initial expansion of stage IB oocytes, the poly-nucleolar cell type responsible for zebrafish feminisation. Given that fem-rDNA overlaps with the only zebrafish locus identified thus far as sex-linked, we hypothesise fem-rDNA expansion could be intrinsic to sex determination in this species.

[1] Department of Anatomy, University of Otago, Dunedin 9016, New Zealand. [2] Department of Biochemistry, University of Otago, Dunedin 9016, New Zealand. Correspondence and requests for materials should be addressed to O.O.-R. (email: oscar.ortega.recalde@postgrad.otago.ac.nz) or to T.A.H. (email: tim.hore@otago.ac.nz)

The germline is established during early development of almost all metazoans with the founding of primordial germ cells (PGCs)[1]. Despite the importance of these in the animal life cycle, the mechanism of PGC specification is not universal for all species. In mammals and urodele amphibians, signal induction from extraembryonic tissue reprograms epiblast cells to PGCs; a process known as epigenesis or induction. In contrast, most other vertebrates including reptiles, fish, birds, and anuran amphibians have an 'immortal' or 'preformed' germline whereby PGCs are specified by cytoplasmic determinants called germplasm. Germplasm is mitotically-inherited from the egg and is asymmetrically segregated during early stages of development[2]. Germplasm components repress somatic differentiation programmes activated in adjacent cells, mediating early germline commitment and helping preserve their developmental potency[3].

The process of PGC specification has been particularly well described in mammals. One of the most astonishing features of PGC specification in mammals is simultaneous genome-wide erasure of DNA methylation marks. In mice, global CG methylation decreases from 71% at PGC specification around day 6.5 (E6.5) to approximately 14% and 7% in male and female at E13.5 PGCs, respectively[4]. DNA demethylation in human PGCs shows a similar dynamic—PGC specification occurs around E12–E16 and DNA methylation drops to approximately 4.5% by week 7[5]. Global methylation reprogramming in marsupial mammals apparently occurs between 10 and 200 days post-partum, when PGCs are well-established in the gonad[6].

Global DNA demethylation in the mammalian germline occurs in sexually undifferentiated PGCs, and is essential for safeguarding against precocious germline differentiation[7,8]. A consequence of extensive erasure in the germline of mammals is that acquired DNA methylation is very rarely inherited[9,10]. Indeed, retention of epigenetic memory in the mammalian genome appears to be largely restricted to imprinted genes and methylated repeats[4]. Of the latter, many appear to be from the intra-cisternal A-type particle class, including those implicated in the transmission of the Agouti viable yellow mouse phenotype[11].

Currently, it is not known if germline erasure of DNA methylation is universal amongst vertebrates, or if it is restricted to species with an induced germline. Indirect evidence suggests that in at least some fish species, epigenetic marks are not erased and can be inherited from one generation to the next. For example, stable silencing of a methylated enhanced green fluorescent protein (EGFP) transgene (GAL4-VP16,UAS:EGFP), is heritable in zebrafish and correlates with DNA methylation levels[12]. In the half-tongue sole (Cynoglossus semilaevis), genetically female fish with ZW sex chromosomes can switch to a phenotypically 'pseudo-male' state (ZWm) by exposure to high temperatures during the juvenile phase. Strikingly, pseudo-males possess high levels of genomic methylation which are inherited to offspring, giving rise to further pseudo-males even without temperature stimulus[13]. These results suggest epigenetic erasure between generations is not prevalent in fish, yet, experiments have not been conducted to test this hypothesis. DNA methylation was not found to undergo extensive erasure in whole zebrafish embryos immediately after fertilisation[14–16], however, these experiments did not involve germline isolation and only sampled the first few days of development.

In addition to being the conduit for inheritance between generations, the germline has been identified as a driver of sex determination in several fish species, including zebrafish[17]. Specifically, zebrafish develop a 'juvenile ovary' around 11–21 days post-fertilisation (dpf)[18–20]. In fish with reduced numbers of germline cells, oocytes undergo apoptosis and male differentiation occurs. In contrast, greater germline cell numbers promote continued female development. Although environmental triggers,

rearing density and small molecules targeting epigenetic modification can influence this process, the primary driver of differential germline proliferation in females and males remains elusive[21–24].

Here, we use a low-cell number bisulfite sequencing pipeline to assess the DNA methylation dynamics in the zebrafish germline. In contrast to mammals, we do not observe genome-wide methylation erasure at any germline stage from 24 h post fertilisation (hpf) until sexual maturity. In addition, we find amplification and demethylation of an 11.5-kb region located in the major sex-linked locus. This region encodes for a type of female-specific ribosomal RNA expressed in oocytes (fem-rDNA) and may play a role in oocyte survival and proliferation. These results provide evidence that the preformed zebrafish germline does not erase epigenetic memory in the form of DNA methylation, and suggests fem-rDNA amplification is implicated in sex determination.

## Results

**Isolation of zebrafish germline cells and low-coverage WGBS.** To obtain germline cells from zebrafish we used the transgenic line Tg(vasa:EGFP)[25]. The reporter gene for this line contains the promoter region of vasa, an RNA binding protein component of the germplasm and well-described germline marker[26]. As such, vasa:EGFP protein is expressed in oocytes and segregated with PGCs during embryogenesis. At 24 hpf, when PGC migration is finished, we found a compact cluster of cells between the yolk ball and yolk extension in the gonadal region (Fig. 1a–d). Given there are few germline cells per individual fish at this developmental stage, ten fish were pooled for each replicate, dissociated with trypsin and prepared for cell sorting. The EGFP +ve cells were isolated with fluorescence-activated cell sorting (FACS) and accounted for approximately 0.01% or less of all cells analysed (Fig. 1e). The gating strategy is exemplified in Supplementary Fig. 1. This percentage is similar to values previously reported for teleost species (0.02–0.04%)[27]. To determine the purity of the population isolated, sorted cells were visualised under an inverted fluorescent microscope. The proportion of EGFP +ve cells ranged from 93.8 to 100% and resembled PGCs in terms of size and shape (Supplementary Fig. 2).

In order to maximise the number of samples tested, we used a low coverage whole genome bisulfite sequencing (WGBS) pipeline to uncover genome-wide methylation levels[28,29]. To be sure we sampled sufficiently, the number of CG calls required to accurately predict global methylation was undertaken using empirical bootstrap sampling[30] of previously published datasets[15]. As expected, we found that increasing the number of CG calls reduces the margin of error for global methylation (Supplementary Fig. 3A). However, beyond a certain threshold, we found increasing the number of CG calls had a minimal effect reducing the margin of error. An asymptotic model described by the equations $y = 1.207/\sqrt{x}$ and $y = 2.109/\sqrt{x}$, for sperm and muscle respectively, was used to fit a curve to the data. At our minimum sequencing depth of 10,000 CG calls, bootstrap sampling predicts a margin of error (99% confidence interval) of approximately ±1.2–2.1 methylation percentage points (Supplementary Fig. 3B).

**Zebrafish germline preserves global DNA methylation.** In mice, epigenetic reprogramming of PGCs occurs in two sequential steps, the first during PGC expansion and migration to hindgut endoderm, the second upon entry of PGCs into the gonads[4]. In marsupials, epigenetic reprogramming occurs postnatally when PGCs have finished migration to the gonad[6]. To capture the full spectrum of these reprogramming windows we measured

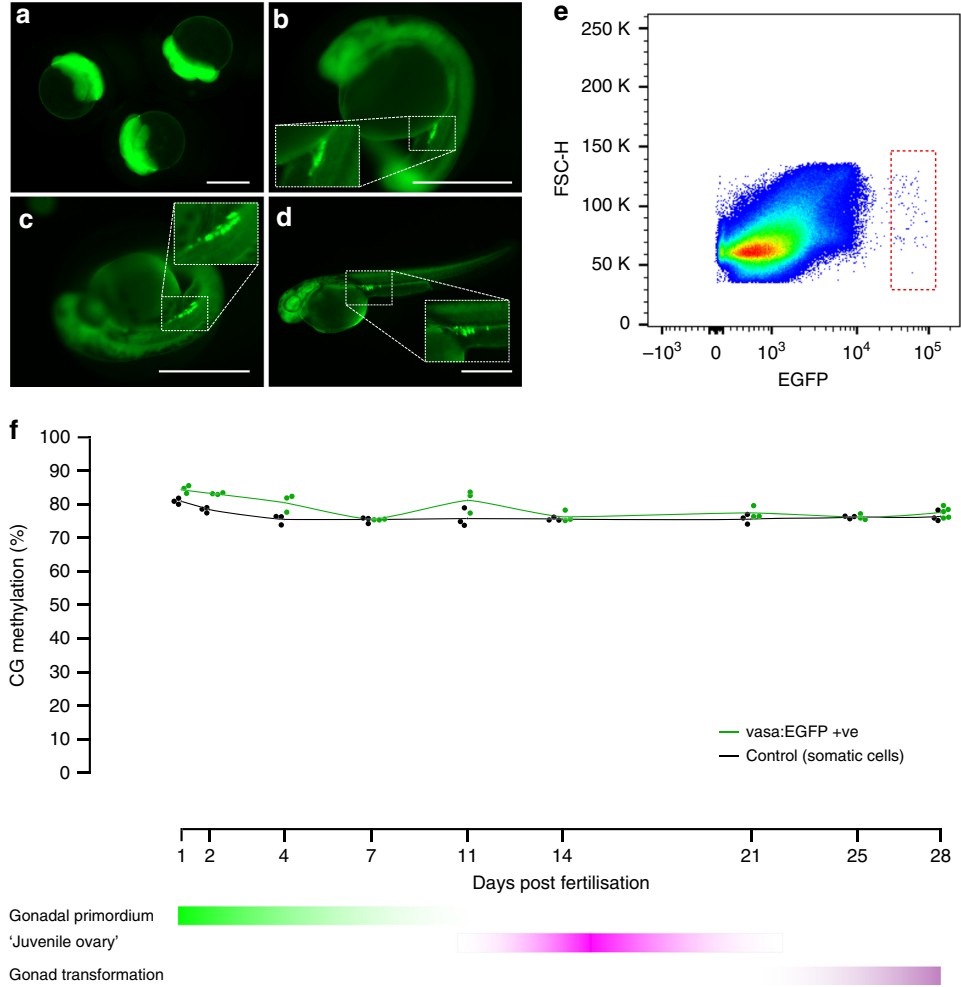

**Fig. 1** Isolation and quantitation of DNA methylation in the zebrafish germline. **a–d** Fluorescence microscopy of *tg(vasa:EGFP)* zebrafish embryos and larvae. 1.5 h post-fertilisation (hpf) (**a**), 24 hpf (**b**), 48 hpf (**c–d**). 1.8× view of EGFP +ve labelled cells is shown inset (dashed lines). Scale bars are 500 µm. Forward scatter height (FSC-H) **e** Flow cytometry plot of 10 zebrafish larvae at 48 hpf. The red dashed square indicates the EGFP +ve population gated for isolation. Blue dots indicate discrete data points (i.e., cellular events), whereas green, yellow and red colouring indicate increasing data density. **f** Percentage of methylation in CG context from 1 to 28 days post-fertilisation (dpf) in both vasa:EGFP +ve germline cells (green line) and control cells (black line). For each sample type and timepoint, $n = 3$ independent biological replicates were used, except for 28 dpf vasa:EGFP +ve, which has $n = 5$ independent biological replicates

methylation from 24 hpf until gonadal transformation, 25–28 dpf. At 24 hpf, EGFP +ve cells were slightly more methylated than control cells, however, both showed high levels of CG methylation (84.06% and 81.41%, respectively). Thus, in stark contrast to mice which experience a massive loss in CG methylation, zebrafish PGCs have preserved global CG methylation upon arrival in the gonad.

Next, we measured methylation levels through gonadal primordium (2–11 dpf), 'juvenile ovary' (11–21 dpf) and early gonad transformation (25–28 dpf) stages. Genome-wide erasure of DNA methylation was not present at any of the time points assessed (Fig. 1f). Average levels of 5-methylcytosine (5-mC) in the CG context were 78.42% and 76.08%, respectively, for EGFP +ve and control cells. Detailed sequencing results are provided in Supplementary Data 1.

**DNA methylation levels during gonad transformation**. Mature germ cells in zebrafish possess sex-specific methylation programmes. In sperm, nearly 95% of CG dinucleotides are methylated, while oocytes are 75% methylated[14,15]. To explore the onset

of this differentiation, we analysed germline methylation from gonad transformation until early gametogenesis. From 25 to 55 dpf, zebrafish gonads undergo an ovary-to-testis transformation in males or further ovarian maturation in females[19,31]. The vasa protein is expressed in male and female germline stem cells[32], yet the intensity of vasa:EGFP expression is correlated with the number of oocytes and can be used to distinguish presumptive females from males[33]. Accordingly, prior to sexual differentiation at 21 dpf, embryos retained a 'juvenile ovary' with low levels of fluorescence detected. At later stages, presumptive male gonads retained low fluorescence whereas female gonads displayed intense fluorescence (Fig. 2a–f).

We isolated germline cells from individual females and males at four time points during gonad transformation (35, 40, 45, 50 dpf). Despite the high expression of vasa:EGFP in mature oocytes, cell filtering prior to FACS restricted cell size to 40 µm. Thus, we were able to collect germinal stem cells (GSCs), oogonia, and stage IA and early IB oocytes[34]. For males, vasa:EGFP expression decreases as germ cells progress through gametogenesis[33], meaning just GSCs and cells during early male gametogenesis were assessed. The male and female germline cells we tested were

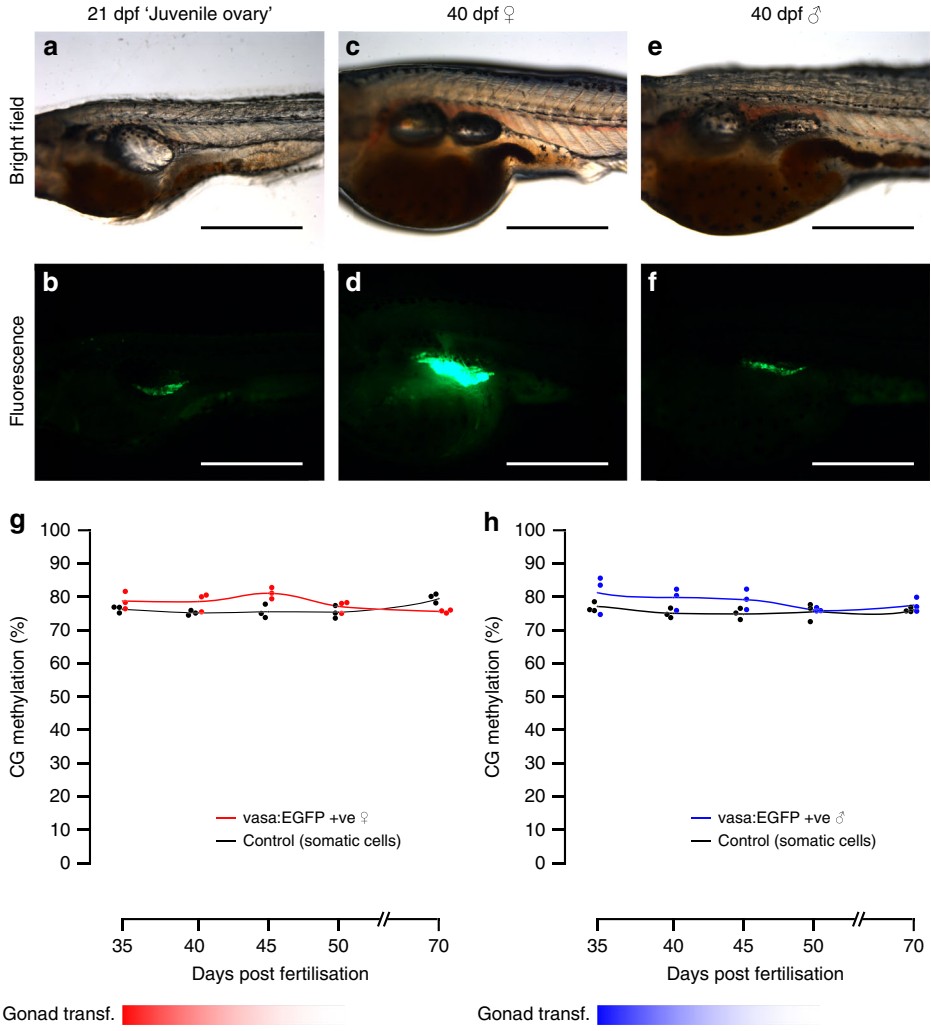

**Fig. 2** Fluorescence microscopy of germline cells and their methylation during gonad transformation. **a–f** Phenotypic sex in zebrafish can be identified using vasa:EGFP expression: during the 'juvenile ovary' stage, expression of EGFP is low but consistent between individuals (**a**, **b**). Later, expression of EGFP vastly increases in presumptive females (**c**, **d**) relative to presumptive males (**e**, **f**). This enables sex phenotyping in early stages of sexual differentiation. Scale bars are 500 µm. **g**, **h** Methylation in the vasa:EGFP +ve germline cells of female (**g**) and male (**h**) fish from the gonad transformation stage until sexual maturity (35–70 dpf). Non-germline control cells were also tested (black lines and dots). For each sample type and timepoint, $n = 3$ independent biological replicates were used

not globally differentially methylated in any of the stages evaluated. The average levels of methylation in CG context for EGFP +ve cells were 78.9% and 79.06%, respectively, for presumptive females and males. Furthermore, global differences in methylation were not found in sexually mature individuals (70 dpf) with average 5-mC levels of 75.64% for females and 77.51% for males. This suggests that hypermethylation of the male germline relative to the female germline, occurs during the final stages of spermatogenesis.

Our low-coverage analysis cannot quantify methylation at single-copy loci, yet, we were able to analyse some genomic subsets. On average 54.31% ($n = 116$, ±SD 1.01) of CG calls mapped to repeated regions (very similar to the overall repeat level of 52%, from Howe et al.[35]). Not surprisingly, we found repetitive regions had greater methylation (mean 86.94%, ±SD 1.88) for all the samples compared to non-repetitive sequences (mean 67.64%, ±SD 4.29) ($p < 0.01$, Wilcoxon signed-rank test). Importantly, there was no hypomethylation of germline samples in either repetitive or non-repetitive subsets relative to non-germline controls (Supplementary Data 1).

**Amplification and demethylation of oocyte-specific rDNA.** Our initial analysis of germline methylation was performed using non-overlapping sliding windows (Figs. 1, 2), however, when all mapped reads were analysed irrespective of their location, methylation levels were greatly reduced for 9 EGFP +ve samples at 28–50 dpf (Supplementary Data 1). One explanation for this was that a lowly methylated region (or regions) was over-represented in our dataset for either technical or biological reasons. When we measured the occurrence of mapped CG calls within sliding 1 Mb windows throughout the genome, we found that the tip of chromosome 4 (Chr4:77,000,001–78,000,000; GRCz11) possessed a surprisingly high density of CG calls (Fig. 3a). Closer inspection revealed that the over-represented reads mapped to both strands of a 17.3-Kb region (chr4: 77,549,891–77,567,278) containing an 11.5-Kb repeat unit encoding 45S ribosomal DNA (chr4:77,555,720–77,567,278) (Fig. 3b). It has been recently reported that at least 2 clusters of rDNA exist within the zebrafish genome[36]. One of these clusters contains the canonical rDNA expressed in all somatic cells. The other cluster is a maternal-specific rDNA type, which we term

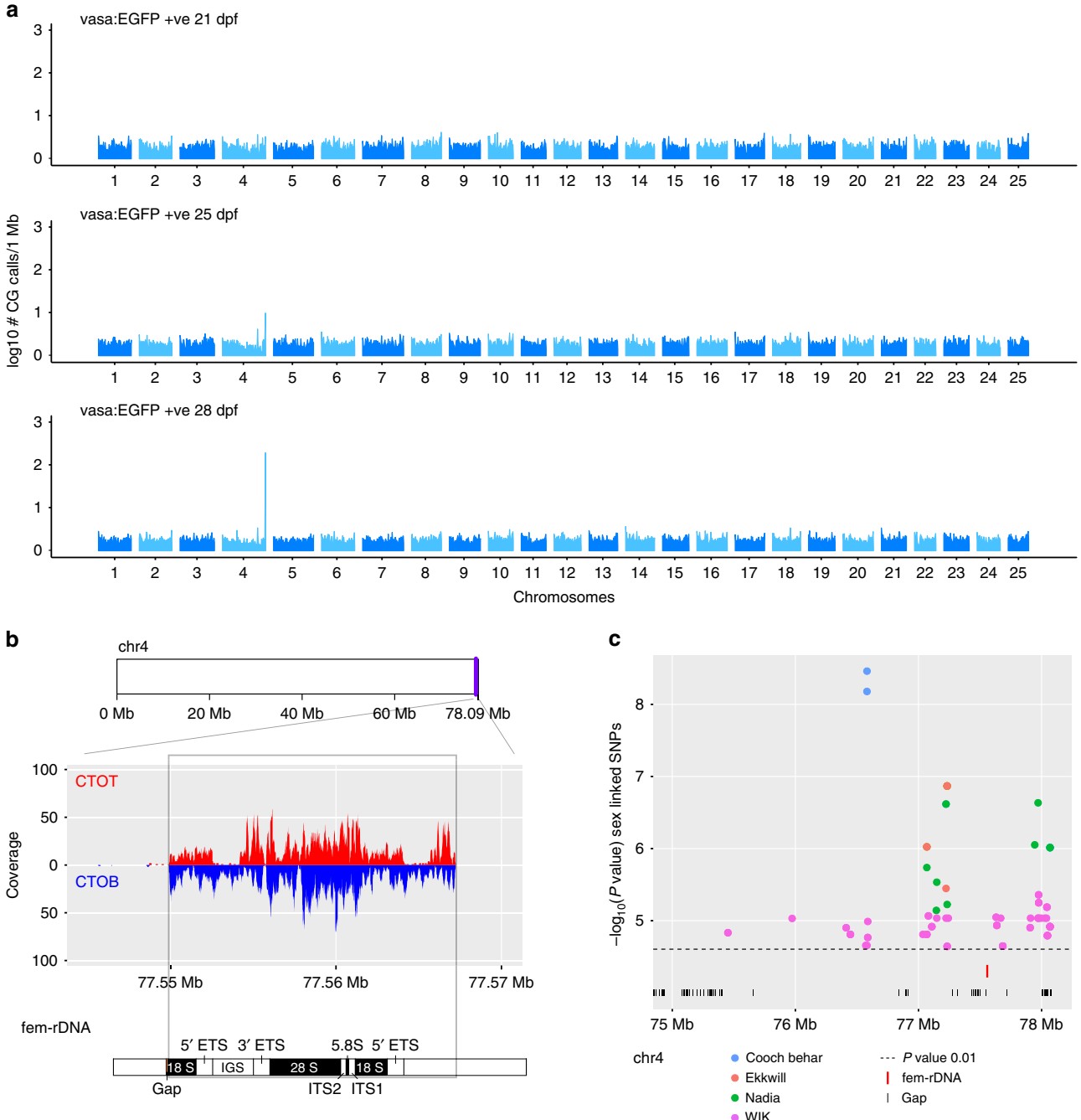

**Fig. 3** Amplification of oocyte-specific fem-rDNA in a previously identified sex-linked region. **a** Number of CG calls mapping to windows of 1 Mb in the whole zebrafish genome. A peak is observed at the right tip of chromosome 4 following 25-dpf and 28-dpf. **b** Reads map to both the complementary original top (CTOT, red) complementary original bottom (CTOB, blue) strand of the 45S fem-rDNA unit on chromosome 4. Components of the rDNA repeat are indicated (External transcribed spacer (ETS), Internal transcribed spacer (ITS), Intergenic spacer (IGS)). **c** The amplified region is located within the most significant sex-linked SNPs from non-domesticated zebrafish strains, Cooch Behar, Ekkwill, Nadia and WIK

fem-rDNA, which overlaps with our overrepresented reads on chromosome 4. The genome reference has assembled only one unit of likely several fem-rDNA copies, separated by intergenic spacers[37].

The tip of chromosome 4 in zebrafish is notable for its close linkage to a previously identified locus associated with sex in natural strains[38]. Analysis of SNPs with the strongest statistical support for sex-linkage in two natural laboratory strains (WIK and EKW) and two recently-sourced wild isolates (Nadia and Cooch Behar) revealed that amplified 45S fem-rDNA is located within the major sex-determining region in chromosome 4

(Supplementary Data 2). The high and variable number of rDNA copies and its location in a poorly-assembled section of the genome makes it difficult to establish the true length of the fem-rDNA repeat. Nevertheless, sex-linked SNPs located at both ends of this gap suggest the complete fem-rDNA cluster is embedded within the sex-determining region (Fig. 3c).

Using our low-coverage BS-seq data, we measured fem-rDNA amplification and methylation levels in both germline and control samples from females, males and sexually indeterminate fish. In the non-germline control samples, we found that fem-rDNA reads, on average, comprised 0.032% of total reads sequenced

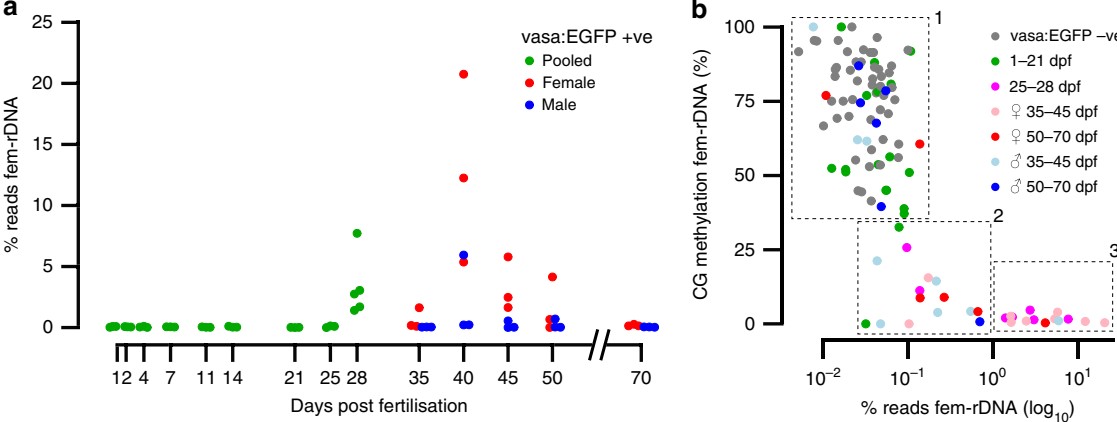

**Fig. 4** Amplification and methylation of oocyte-specific fem-rDNA during gonad transformation. **a** Percentage of reads mapping to fem-rDNA in the germline prior to sexual differentiation (green), and in presumptive males (blue) and females (red) during gonad transformation and after sexual maturation. **b** Relationship between the amplification and methylation of fem-rDNA for vasa:EGFP −ve control samples (grey dots), and vasa:EGFP +ve germline cells from sexually undifferentiated fish at 1–21 (green dots) and 25–28 dpf (magenta dots); presumptive female fish at 35–45 (pink) and 50–70 (red) dpf; presumptive male fish at 35–45 (light blue) and 50–70 (blue) dpf. Samples have been divided into 3 clusters based on rDNA level and methylation; (1) 'background' non-amplified and methylated (2) moderately amplified and lowly methylated, (3) highly amplified and unmethylated. These consist of $n = 75$, $n = 15$, and $n = 14$ samples, respectively

(Supplementary Data 3). These levels were similar to the proportion of fem-rDNA reads in adult muscle tissue from other datasets (0.049%, Supplementary Data 4)[15]. As such, we considered this range to represent non-amplified, background levels of fem-rDNA. Fem-rDNA levels were also at the background level for 41 out of 55 germline samples but there was clear enrichment for fem-rDNA in 8 females and one presumptive male (>1% reads mapping to fem-rDNA), peaking at 40 dpf (Fig. 4a). For one female, 20.75% of reads mapped to fem-rDNA, representing amplification of at least 170-fold compared to non-germline controls. To validate this finding, we performed quantitative PCR using somatic-rDNA as a similar multi-copy genomic control and independently grown fish. We found fem-rDNA was amplified relative to non-germline cells in 9 out of 12 female germline samples (>5-fold) at 35 and 40 dpf (range 5.89-93.54-fold amplification relative to background), but found no amplification in a further three males (Supplementary Fig. 4).

When we analysed fem-rDNA methylation, we found three clear groups. All non-germline controls and many male and female germline samples were highly methylated (mean 73.2%, ±SD 17.29%), and did not show fem-rDNA amplification (mean 0.04%, ±SD 0.02%, see group 1, Fig. 4b). In contrast, those with strong amplification of fem-rDNA (i.e., >1% of total reads) were fully demethylated (mean 1.72%, ±SD 1.29%), and except for one individual, were either phenotypically female or late in the sexually indifferent phase (see group 3, Fig. 4b). An intermediate group of germline-only samples showed modest amplification of fem-rDNA (0.032-0.703% reads mapping to fem-rDNA, mean 0.23%, ±SD 0.22%) and were lowly methylated (mean 10.08%, ±SD 10.1%). Together this shows that fem-rDNA amplification and demethylation is highly correlated with feminisation of the zebrafish gonad.

## Discussion

DNA methylation represents a stable yet flexible gene expression control system that is critical for formation of cell identity during development[39,40]. In mammals, global erasure of DNA methylation is closely related to the acquisition of developmental potency in the early embryo and during re-animation of the bipotential germline during PGC definition (reviewed in ref. [41]).

In species with a preformed germline, where PGC specification relies on heritable maternal factors and not dedifferentiation of somatic cells, existence of epigenetic erasure and reprogramming is unknown. In this study, we employed low coverage WGBS-seq to evaluate DNA methylation dynamics in the zebrafish germline throughout development. In stark contrast to mammals, we find DNA methylation is not erased at any stage of germline development ranging from 24 hpf until sexual maturity. While we could not test germline cells for demethylation less than 24 hpf (oocyte-derived vasa:EGFP is found in somatic tissues at this time), Skovortsova et al.[42], in this issue, isolated germline cells from multiple time-points less than 36 hpf using an alternative transgenic line, and also found no global reprogramming. Indeed, compared to somatic cells, deep-sequencing of PGCs by Skovortsova et al[42]. revealed very few unique germline methylation patterns, despite markedly divergent transcription.

In mammals, epigenetic memory in the form of CG methylation is carefully maintained at 70–85% in adult somatic tissues, with significant demethylation only occurring in pathological situations[43,44]. Global DNA demethylation in the early embryo is tightly linked to acquisition of naïve pluripotency[45–47], with a second and more dramatic erasure event occurring in PGCs, where reprogramming helps activate the germline programme[8,48]. In contrast to mammals, species with a preformed germline such as Xenopus and zebrafish do not require de novo formation of PGCs at each generation and instead use inherited cytoplasmic determinants to continuously define germline cells[49]. In line with the lack of cellular reprogramming required, our study shows that global DNA methylation erasure is not a feature of germline specification in zebrafish (Fig. 5a). Given the vast majority of vertebrate species similarly define PGCs using this mechanism[49], it seems likely that bulk intergenerational preservation of DNA methylation exists in other non-mammalian vertebrates, but this remains to be tested. The absence of DNA methylation erasure at early zebrafish embryo stages[14–16] and in the germline (this study and Skovortsova et al.[42]) provides a mechanistic explanation through which DNA methylation at transgenes can be stably inherited between generations[12]. While transgenerational epigenetic inheritance appears to be a rare (but potentially important) mode of mammalian inheritance[9,10], our data suggest adaptive epigenetic changes in response to

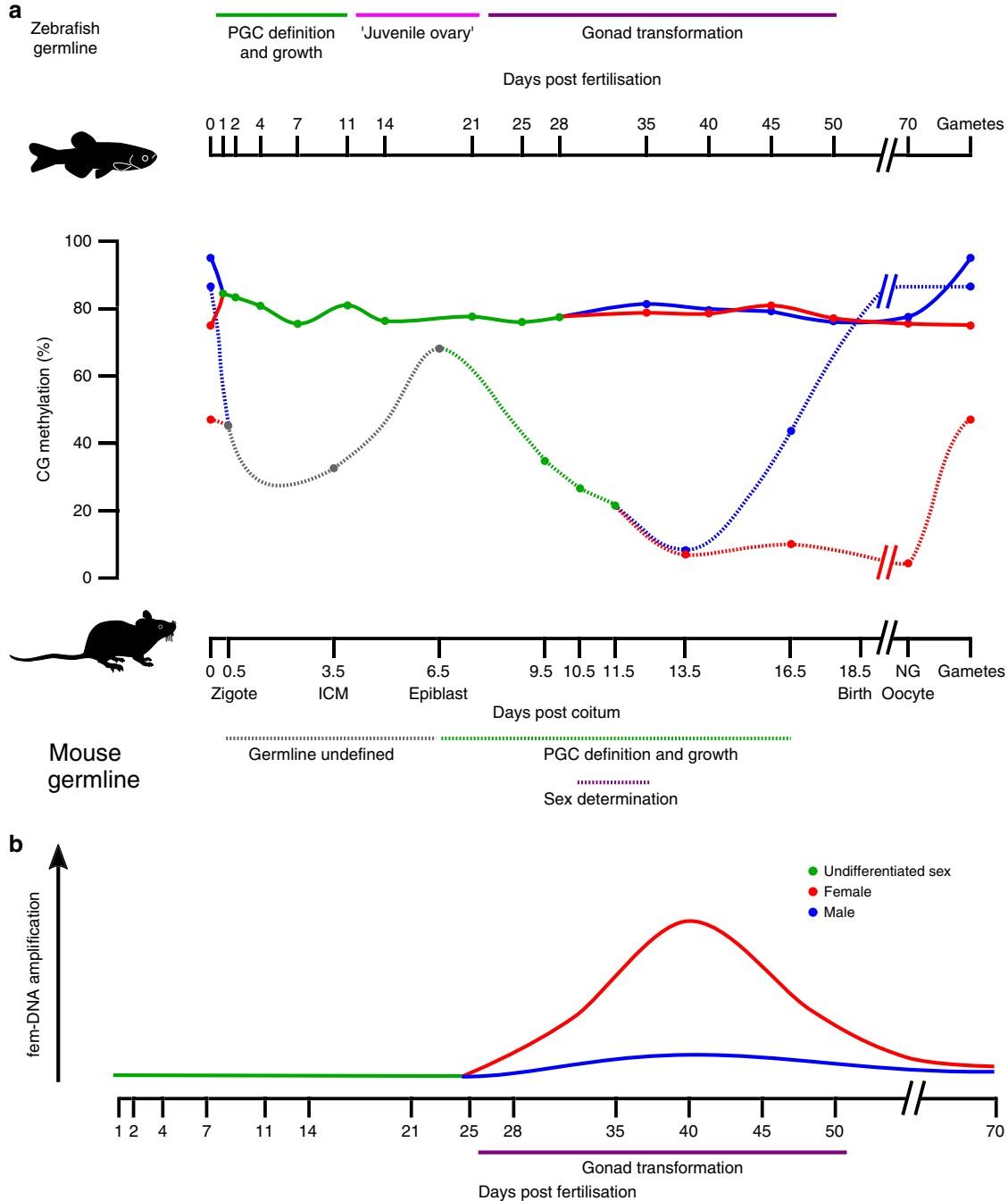

**Fig. 5** Global DNA methylation and fem-rDNA dynamics in the zebrafish germline. **a** CG methylation in the mouse and zebrafish germline. In stark contrast to mice, the zebrafish germline does not undergo extensive erasure of DNA methylation and germline DNA from females (red), males (blue) and fish of undifferentiated sex (green) are similar. Note, data for mouse were taken from the meta-analysis provided by Lee, Hore and Reik[41]. **b** Striking amplification of fem-rDNA occurs in germline cells during the critical period of gonad transformation in females. Beginning around 28 dpf perinucleolar oocytes amplify in at least 2 orders of magnitude oocyte-specific 45S rDNA. ICM, inner cell mass; non-growing (NG) oocyte

environmental cues may be comparatively more significant in non-mammalian vertebrates.

Sexually dimorphic DNA methylation emerges early during gonadal development in mammals. For mice, global methylation levels in PGCs at day 13.5 post-coitum, soon after the beginning of sex differentiation, drop to 7% in females and 14% in males[4]. The oocyte epigenome remains significantly hypomethylated in relation to sperm in mice being 40.0% and 89.4%, respectively[50]. In zebrafish global levels of methylation in sperm are 95% whereas mature oocytes are 20% lower[14,15]. These gamete-specific

methylation patterns appear to be generated relatively late in development compared to mice (Fig. 5a)—we find consistent methylation levels in both the male and female zebrafish germline from the gonad transformation period until the point of sexual maturity for cells in the early stages of gametogenesis. Isolation of male germline cells from late gametogenesis, using either single cell analysis or a marker more versatile than vasa:EGFP[51], will refine our understanding of when gross sexually-dimorphic methylation patterns become apparent in zebrafish. In addition, full-coverage sequencing during the gonad transformation period

may uncover sex-specific methylation at single-copy loci, however, the fact that global methylation remains similar between sexes supports previous claims that germline stem cells remain plastic in adults and can rapidly switch to producing the alternative gamete[52].

Extrachromosomal amplification of rDNA by rolling circle intermediates has been described in several animal species including *Xenopus* and zebrafish[36,53–55], and is thought to primarily help support the unique metabolic demands of the oocyte. Our study used detailed quantitative sequencing techniques in the germline to uncover >170-fold amplification of maternal-specific 45S rDNA (called here 'fem-rDNA'). We found fem-rDNA first appears in the germline at 28 dpf (Figs. 4a, 5b). At this stage, perinucleolar oocytes become particularly abundant in presumptive females[20,31] and are characterised by the proliferation of nucleoli, a structural manifestation of ribosome biogenesis[19,56]. Perinucleolar oocytes are critical for sex determination in zebrafish; as the central signalling cell responsible for feminisation of the gonad, decreased numbers of perinucleolar oocytes lead to a male gonadal fate[20,22,57]. By definition, an inability to amplify fem-rDNA would block formation of perinucleolar oocytes, and presumably therefore, suppress female differentiation. Interestingly, we found one presumptive male showed fem-rDNA amplification. The ovary-to-testis transition in male zebrafish involves degradation of perinucleolar oocytes[33], a process potentially still ongoing in this individual.

In addition to creating the defining characteristic of perinucleolar oocytes, the genomic location of fem-rDNA suggests it may be implicated in sex determination. A 1.5-Mb region overlapping fem-rDNA is strongly linked to sex phenotype in non-domesticated zebrafish strains[38], and as yet, no candidate sex-determining genes have been identified within it. Sex-linked machinery or regulatory elements that control fem-rDNA transcription or amplification, or even differential fem-rDNA repeat number, could contribute to sex determination in wild strains of zebrafish.

In domesticated zebrafish strains, where no regions of the genome appear to be sex-specific, non-genetic factors may contribute to rDNA amplification or transcription. Accessibility and transcription of ribosomal genes is strongly associated with epigenetic regulation. For example, loss of methylation in the spacer region of rDNA is inextricably linked with transcriptional activation in *Xenopus*[58], and methylating a single CG within the mouse rDNA promoter represses transcription in vitro[59]. It is tempting to speculate that epigenetic modification of fem-rDNA may help explain skewed or non-mendelian sex ratios in domesticated zebrafish strains. While this possibility remains to be tested, it is intriguing that the demethylating agent 5-azacytidine induces feminisation of zebrafish, as would be predicted if epigenetic modification of fem-rDNA played a central role in sex determinaton[24].

In conclusion, our work demonstrates (i) the absence of global DNA methylation erasure in the zebrafish germline and (ii) extensive amplification and demethylation of the oocyte-specific fem-rDNA cluster during gonad transformation. By showing that epigenetic memory in the form of DNA methylation is not erased in the germline from 24 hpf until sexual maturity, we provide a mechanistic explanation for transgenerational epigenetic inheritance in species with a preformed germline and suggests DNA methylation therefore may have an underappreciated role in heredity and evolution. In addition, the amplification and demethylation of fem-rDNA in peri-nucleolar oocytes, the key cell type signalling feminisation of the zebrafish gonad, suggests fem-rDNA has a critical function in sex determination for this species.

## Methods

**Zebrafish husbandry and collection.** Use of zebrafish in this study was approved by University of Otago Animal Ethics Committee (ET 25/2017). Adult *Tg(vasa: EGFP)* zebrafish[60] were maintained under standard conditions at the Otago Zebrafish Facility, University of Otago[61]. Embryos were obtained through natural spawning and grown in 28.5 °C egg water (NaCl 5.0 mM, KCl 0.7 mM, CaCl 0.33 mM, MgSO$_4$ 0.33 mM). After the hatching period, larvae were transferred to the central system. Embryos, larvae, young and adult fish were euthanized by rapid cooling in ice cold water for 10 min[62] and then they were visualised with a LEICA M205 FA fluorescence microscope and a LEICA DFC490 CCD camera.

**Preparation of embryonic cells for sorting.** Fish of different developmental stages were dissociated by vigorous pipetting in 500 μl TrypLE™ Express (ThermoFisher, 12604021). For 24-hpf and 48-hpf embryos, eggs were manually dechorionated using two tweezers, for post-hatching fish until 14 dpf whole fish were trypsinized. For older fish (>14 dpf) the gonadal region was dissected, and cells were dissociated. To stop the trypsin reaction, and stain cell nuclei, we added 20 μl of foetal calf serum plus 1:10,000 DAPI (ThermoFisher, 10091-148) per reaction. Disaggregated cells were separated from debris using a 40 μm nylon cell strainer (Biologix, 15-1040) and keep on ice prior sorting.

**Fluorescence-activated cell sorting of zebrafish germ cells.** Disaggregated cells were passed across a 488 argon laser to detect EGFP (BD Fortessa, BD Biosciences; BD FACSAria sorter, BD Biosciences). Forward scatter (FSC) and side scatter (SSC) parameters were used for observation of the cell distribution profile. We used relative FSC, SSC, and EGFP intensities to identify a germline subpopulation and these cells were gated and sorted. Cells were collected in 0.2 ml tubes containing 20 μl ddH$_2$O and stored at −80 °C or in 100 μl of PBS for fluorescence visualisation. For methylation analysis of embryos, larvae and juvenile fish between 1 and 28 days, 3 replicates for EGFP −ve and +ve were obtained for each time point (1, 2, 4, 7, 11, 14, 21, 25, 28 dpf), except for EGFP +ve cells at 28 dpf when 5 replicates were analysed (*n* = 56). To evaluate the gonad transformation period and sexual maturity (35, 40, 45, 50, 70 dfp), 3 replicates were obtained in a similar fashion for each sex (*n* = 60). As such, the total number of samples purified for high-throughput methylation analysis was 116. Additionally, EGFP +ve and −ve cells were sorted from 15 individuals between 35 and 40 days; samples which were used to verify fem-rDNA using quantitative PCR. The purity of sorted cells was assessed using raw images captures by an IncuCyte FLR imaging system (Essen Instruments).

**DNA extraction.** Total nucleic acids were purified using the Bio-On-Magnetic-Beads (BOMB) approach[63]. Briefly, a guanidine isothiocyanate lysis buffer was used to homogenise cells and then was combined with TE-diluted Sera-Mag Magnetic SpeedBeads (GE Healthcare, GEHE45152105050250) and isopropanol in a volumetric ratio of 2:3:4 (beads:lysate:isopropanol). Beads were captured with a neodymium magnet and washed once with isopropanol, twice with 70% ethanol and resuspended in milliQ water.

**PBAT library preparation and sequencing.** Bisulfite-converted genomic libraries were prepared using a modified post-bisulfite adaptor tagging (PBAT) method[64,65]. Bisulfite treatment was performed according to the EZ Methylation Direct Mag Prep kit (Zymo, D5044) instruction manual. To synthesise the first strand, we used converted DNA and 5′-biotinylated adaptor primers bearing seven random nucleotides at its 3′ end (BioP5N7, biotin- ACACTCTTTCCCTACACGACGCT CTTCCGATCTNNNNNNNN). The first strand product was purified using streptavidin-coated magnetic beads (ThermoFisher, 11205D) and alkaline denaturation. Second strand DNA was synthesised using the immobilised first strand DNA and another adaptor primer also bearing seven random nucleotides at its 3′ end (P7N7, GTGACTGGAGTTCAGACGTGTGCTCTTCCGATCTNNNNNNNN). Second strand DNA was eluted, amplified by 15 cycles of PCR and size selected by PEG-diluted SPRI beads. During PCR, sample-specific barcodes and sequences required for Illumina flow-cells binding were added to libraries using 1× HiFi HotStart Uracil + Mix (KAPA, KK2801 and 10 μM indexed Truseq-type oligos). Library integrity was assessed by agarose gel electrophoresis and sequenced on a 150 bp single-end run on Illumina MiSeq.

**Bioinformatic analysis.** The quality of the raw FASTQ files was evaluated using FastQC software (v0.11.4). Raw reads were trimmed using Trim Galore! v0.4.2, in a two-step process. First, adaptors were removed and 10 bp was hard-trimmed from the 5′ end of all reads and low-quality base calls (Phred score < 20) were removed. Read mapping and base calling was performed using Bismark v0.19.0[66] with the option --pbat specified. Zebrafish genome version 11 (GRCz11) was used as reference. Global methylation in CG context was calculated as the proportion of total methylated cytosines in CG context over total cytosines in CG context using non-overlapping windows of 10 Mb in SeqMonk programme v1.43.0. The non-conversion rate during the bisulfite treatment was evaluated by calculating the proportion of non-CG methylation; by this measure, all libraries must have had a bisulfite conversion efficiency of at least 96.02% (Supplementary Data 1). A BSGenome data package was forged for using the latest UCSC zebrafish genome

version 11 (GRCz11) (BSGenome v1.46)[67] and used for all the analysis requiring this dependency in R (v3.4.4). To analyse differential DNA methylation levels in repetitive and non-repetitive sequences we used RepeatMasker annotations obtained from UCSC (v4.0.5 RepBase library: 2014013)[68].

**Margin of error estimation for low-coverage WGBS**. To evaluate the number of CGs required to accurately predict the genome-wide methylation status we used empirical bootstrap sampling[30] from previous high-coverage zebrafish methylation datasets (SRR800056, SRR800081)[15]. Briefly, we used fastq-tool v0.8 (https://homes.cs.washington.edu/~dcjones/fastq-tools/) to obtain 1000 random samples with replacement in regular intervals of CG calls from approximately 100 to 30,000. Each sample was processed as mentioned previously and the proportion of data falling within the 0.5–99.5 percentiles was calculated to generate a margin of error (99% confidence interval).

**Ribosomal DNA bioinformatic analysis**. To determine overrepresented regions among the genome, EGFP +ve replicates for each time point (21, 25, and 28 dpf) were merged and the number of CG calls per Mb was calculated and divided by the average number of calls in all the probes. To identify the amplified region, reads were aligned to complementary converted strands of chromosome 4 (CTOT and CTOB) using Bowtie2 v2.3.2[69] with the --very-sensitive option (-D 20 -R 3 -N 0 -L 20 -i S,1,0.50) to increase mapping sensitivity and accuracy. Manhattan plot and coverage plot were drawn using ggbio v1.26.1[70].

The genome coordinates of sex-linked SNPs previously published[38] were converted from Zv9 to GRCz11 using CrossMap v0.3.0[71]. Previous methyl-seq datasets were obtained from SRA and processed as mentioned above with the option --directional specified[15]. Number of reads and CG methylation for fem-rDNA were quantified for the region chr:77,549,891:77,567,278. For low coverage WGBS, methylation for fem-rDNA was calculated as the proportion of methylated C's in CG context over the total C's in CG context in samples with at least 10 calls for the region of interest. For deep sequenced datasets, fem-rDNA methylation was quantified as the proportion of methylated C's in CG context over the total number of C's in CG context within the 17.3 Kb amplified region. Scatter plots were drawn using ggplot2 v3.0.0 in R v3.4.4.

**Quantitative PCR**. Quantitative PCR was performed using the SensiFAST™ SYBR® No-ROX Kit (Bioline, BIO-98020) and the LightCycler® 480 instrument (Roche). Specific PCR primers sequences for fem-rDNA and somatic rDNA are listed in Supplementary Table 1. Negative controls included EGFP −ve cells from females and males at 35 dpf, EGFP +ve cells from males at 35 dpf and embryonic cells at 24 hpf. The final volume in each reaction was 12 µl including 6 µl of SensiFAST™ SYBR® No-ROX mix and primers at a concentration of 900 nM. The reactions were incubated in white 96-well plates at 98 °C for 5 min, followed by 45 cycles of 98 °C for 20 s, 60 °C for 15 s, and 72 °C for 20 s. All reactions were run in duplicate. Data was analysed with the LightCycler® 480 software (Roche) determining the threshold cycle (Ct) by the second derivative max method. A baseline level of amplification was determined as the mean of Δ Ct (Ct fem-rDNA – Ct som-rDNA) for non-germline samples (EGFP −ve females and males) and the value obtained was used as control for sample normalisation (i.e., ΔΔ Ct method).

**Statistics**. Statistical analysis between groups was performed using Wilcoxon signed-rank tests. In all cases significance was set as $p < 0.05$.

## Data availability

The accession number for the FastQ files and CG calls of the low coverage WGBS libraries reported in this paper is GEO: GSE122695. All other relevant data supporting the key findings of this study are available within the article and its Supplementary Information files or from the corresponding author upon reasonable request. The source data underlying Figs. 1a–f, 2a–h, 3a–c, 4a–b, 5a-b and Supplementary Figs. 1−5 are provided as a Source Data file. A reporting summary for this Article is available as a Supplementary Information file.

## Code availability

The source code of the analysis is publicly available on Github at https://github.com/OscarOrt/Met_zebrafish_germline

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

## Acknowledgements

We thank Michelle Wilson for support and technical assistance in cell sorting; Noel Jhinku for zebrafish husbandry, Parry Guilford for sequencing assistance and Jo-Ann Stanton for quantitative PCR support. This work was supported by the University of Otago Research Grant (111899.01.R.LA) and a Marsden grant (18-UOO-200). O.O.-R. was supported by a PhD scholarship from the University of Otago.

## Author contributions

T.A.H. conceived and funded the study. T.A.H. and O.O.-R. designed the experiments, with contributions from N.J.G. O.O.-R. collected the samples and performed the laboratory work. O.O.-R. and T.A.H. performed bioinformatic analysis. R.C.D. aided with sequencing. O.O.-R. and T.A.H. wrote the manuscript. All authors contributed and approved the final manuscript.

## Additional information

**Competing interests:** The authors declare no competing interests.

