## [Peer Review File · Nature Communications]

Reviewers' comments:

Reviewer #1 (Remarks to the Author):

The study by Ortega-Recalde and colleagues reveals that in contrast to mammalian epigenomes, the germline epigenome is not reprogrammed in zebrafish. This was accomplished by isolation of PGCs and profiling bulk cytosine DNA methylation levels using low coverage WGBS. This result will likely extend to other species that have a preformed germline including birds, amphibians and other fish. They also discovered amplification of the 45S rDNA, which coincided with loss of DNA methylation. Intriguingly, this locus was in a region that is known to be sex-linked.

Overall, this is an interesting and well-written study. The authors are mostly careful not to overstate their results, especially with regards to the amplification of the 45S rDNA. Currently, only an association between 45S rDNA and feminization is reported and there are clear exceptions numerous males were found to have similar molecular phenotypes. Line 19 in the abstract will need to be modified as it is not female specific if it occurs in males as well. Either way, this process is more prevalent in females than males.

I have a few comments that need addressing.

1. I can find no information on the number of samples, replicates or deposited data for this study?
2. Similarly, I can find no information on the bisulfite sequencing conversion rates, the number of sequenced reads and alignment scores. These are especially relevant for understanding if there is potentially a technical reason why the rDNA is being overamplified. The data in Figure 5 could be due to technical artifact given there is so much biological variation in the results. There are males throughout all three sections, even with females there is a broad range of methylation/DNA amplification.
3. The emphasis on transgenerational inheritance throughout the entire manuscript is off-topic. There are no experiments executed to test transgenerational inheritance based on some environmental exposure. If the authors want to mention the possibility of TGI in the discussion this is fine, but there is no need to mention it in the abstract, intro, results, etc.
4. Figure 1 seems better suited as a supplementary figure.
5. Line 107, there is no data to support that the rDNA overamplification "helps orchestrate sex determination". It's purely a hypothesis at this point.
6. Given the use of low coverage WGBS, the authors need to make clear in the discussion that there could be localized differences to methylation that are not detected by this method.

Reviewer #2 (Remarks to the Author):

In the manuscript, "Zebrafish preserve germline epigenetic memory globally but demethylate and amplify sex-linked rDNA during feminisation", Ortega-Recalde et al use low cell number bisulfite sequencing to assess methylation of zebrafish germ cells at a total of 14 stages of development between 1 and 70 days post fertilization. Using this data, they address a longstanding question in the field of chromatin biology/germ cell development, which is whether or not the methylation state of non-mammalian vertebrate germ cells is reprogrammed in the same way it is in mammals. They report that global methylation erasure was not detected at any stage assessed. Analysis of methylation at individual sequences across these time points was not provided with one exception. Very intriguingly, the authors find a region of zebrafish chromosome 4 that contains ribosomal genes and is associated with sex determination undergoes demethylation and amplification in females around 35 dpf. Over all, the paper includes two important findings, that

are likely to be of interest to many. However, both are analyzed at a fairly superficial level.

For the methylation analysis, while it is useful to know that global methylation erasure/reestablishment is not detected at the stages examined, data in figure 1 and table S1 seem to show relatively significant changes in methylation (for example methylation seems to be around 84% at 1 and 2 dpf but at 75% at 7 dpf). It would be useful to know if there are subsets of sequences that change their methylation status between these (or other) stages. The authors also don't comment clearly on how transposons/pericentromeric sequences were handled in their analysis. Were they excluded or included? If excluded, the caveats associated with doing so should be highlighted in the discussion.

The second finding of sex specific demethylation and amplification on chromosome 4 is potentially very exciting, but exploration of this finding feels almost too limited to really appreciate its significance.

First, there is no secondary method of validation for what is a somewhat extraordinary observation. This is needed. Can the authors use Southern blotting to clearly demonstrate amplification of the repeat? (The repetitive nature of the region should counter any issues related to limited sample availability). Other PCR based methods could likely be used to confirm if Southern blots aren't technically feasible due to sample limitation.

Second, while the link between amplification and demethylation and sex determination is tantalizing, the manuscript would be made stronger by further insights, such as additional data addressing mechanisms and /or consequences. A few potential questions that could be addressed are: Does exposure to the demethylating agent 5 azacytidine lead to amplification of this region in germ cells or in other contexts? Is demethylation/amplification triggered by Tet enzymes? Is demethylation and amplification accompanied by other chromatin changes? What does methylation of this region look like in wild zebrafish? Is demethylation of this region required for amplification/transcription/sex determination. Exploring all these directions would likely be beyond the scope of a single manuscript. However, additional insights into some aspect of the biology associated with this finding would help to better place it in context.

Minor:

In Fig 2 It is hard to see the germ cells perhaps a zoom in or higher magnification would be helpful

Personally, I found figure 1 to be fairly confusing, I would suggest omitting.

Reviewer #3 (Remarks to the Author):

General comments

This ms addresses whether vertebrates with a "pre-formed" germline (most species) demethylate DNA during early development as it occurs in mammals and urodele amphibians, which have an "induced" germline and, according to studies in mice, exhibit two waves of global demethylation during early development. Here, using zebrafish they find that there seems to be no demethylation events in the germline.

One of the major problems of this study is that no methylation data is provided within the first 24 h post fertilization. If embryos were collected at 1.5 h post fertilization (hpf; Fig. 2A) why then CG Methylation was measured only starting at 24 hpf? In fact, during the first 24 h after fertilization important events take place and it is likely that by measuring CG methylation only starting at 24 hpf these events may have been missed. One can argue that there are previous studies covering the first 24 h in zebrafish, but this opportunity should be taken to repeat those analysis. Not providing robust data concerning the first 24 h can invalidate the main conclusion of this paper. Thus, samples should be taken also at fertilization and with the first 24 subsequent hours in

addition to other time points.

Specific comments

Between lines 119 and 122 there should be a description on the methods used to sacrifice the fish, the approved protocol, etc. All these details are missing.

Line 178. Why the better assembled version 11 of the zebrafish genome (GRCz11) was not used?

Line 209. "mentioned above", not "mention above".

Line 268. Use "PGC" instead of "primordial germ cells".

Line 342. "It has been recently reported that..." Add citation.

Line 372. "Many germline samples". How many? Were they males or females?

Line 374. "... in some females..." How many? Why not all of them?

End of line 387. Add "(0-30%)".

Line 413. "species with an induced germline" should be "species with a pre-formed germline".

Line 470. The data of this ms does not support the claim that "fem-rDNA" is further implicated in sex determination". At best it shows some sort of correlation, which does not mean causation.

Reviewer #4 (Remarks to the Author):

Ortega-Recalde and colleagues present an interesting study which characterises the DNA methylome during zebrafish germ line development. They do not identify a global DNA demethylation event, as documented in mouse and human – an important and provocative observation. In addition, they present evidence of rDNA amplification in female germ cells –and suggest this might be related to the previously uncharacterised mechanism of sex determination in zebrafish.

Overall this is an interesting and well-conducted study, by a group with a strong pedigree in the DNA methylation field. It will be of general interest, in particular to researchers studying germ line development and epigenetic reprogramming. However, there are some issues that need to be addressed prior to publication.

1. rDNA amplification. The possible connection to sex specification is interesting. While it is not necessary that the authors fully uncover the mechanism, I think further discussion, or a model, of how this might work should be included. If rDNA is mechanistically linked to sex specification in germ cells then would this not lead to a progressive increase in rDNA over generations? How do the authors imagine this works?

2. Epigenetic reprogramming vs DNA demethylation. Throughout the manuscript the authors tend to use these terms interchangeably (for instance ' Results: Line 267 'Absence of genome-wide epigenetic reprogramming in the zebrafish germline'. Repeated line 417). This paper deals specifically with DNA demethylation and so the authors should not broaden their claims. It is possible that there are global changes in other epigenetic marks (histone modifications or histone variants), perhaps in a manner akin to the mouse germline, that occur independently of DNA demethylation in zebrafish. All instances should be corrected, and the authors may wish to discuss this issue.

3. The quantification of DNA methylation is based solely on WGBS analysis. The merits of this as a methodology to assess global DNA methylation levels is disputed. In particular, the presence of repetitive elements with unknown copy number is problematic – and they likely represent a significant proportion of global DNA methylation. Have the authors validated their findings with an orthogonal quantitative method, such as LC-MS?

4. The authors present convincing data that there is no significant global DNA demethylation at the stages analysed. Is it possible that DNA demethylation occurs at a later or (perhaps less likely) an earlier stage? For instance,

- given the low level of 5mC in eggs, presumably there must be DNA demethylation at some point during oogenesis? When does this occur?

- Might it be that DNA demethylation occurs in sex specific manner in zebrafish? This may be different in zebrafish due to the apparently enigmatic mechanism of sex determination.

- is it possible that there is also a period late DNA demethylation followed by DNA remethylation in the male germ line? Can the authors formally rule this out?

- alternatively, is it possible that there is early DNA demethylation and subsequent remethylation that occurs in the earliest specified PGCs?

5. If there is no reprogramming of DNA methylation, is there evidence of more variability in DNA methylation patterns between zebrafish individuals (as compared to mice). Would the authors expect differences to emerge in different populations (or across generations), as any sporadic accumulation of DNA methylation would not be reset in the germ line? Might this impact the CG content in the genome over evolutionary time?

6. Introduction:

Page 2

- PGCs are not stem cells. They do not self-renew
- the cells from which PGCs are induced are not somatic, they are pluripotent progenitors.

Page 5

– Abstract and introduction: ‘no barrier to transgenerational epigenetic inheritance’. This is not necessarily the case. There may be locus specific mechanisms to erase DNA methylation (i.e. some barrier, albeit not complete). Erasure of DNA methylation may occur at a different stage (see point 4). There may also be erasure/reprogramming of other marks. On a related point, in mammals there is little evidence that DNA methylation leads to transgenerational epigenetic inheritance in any meaningful way (even when it escapes reprogramming). Other mechanisms have been proposed (such as tsRNA) - so there may be barriers to such mechanisms in zebrafish PGCs.

7. The section 412 – 419 is confused/confusing.

- xenopus and zebrafish do not have an induced germline
- commitment is a strange term here.
- prominent – possible maybe?
- it is worth noting that although preformation is more common (likely because it encourages speciation), epigenesis is the conserved mechanism

8. Page 21: line 411. reference. ‘loss of 5methyl-cytosine in the early embryo is tightly linked to acquisition of naïve pluripotency’. The primary papers should be cited here Habibi et al., Leitch et al., Ficz et al. – all 2013.

Reviewer #1:

The study by Ortega-Recalde and colleagues reveals that in contrast to mammalian epigenomes, the germline epigenome is not reprogrammed in zebrafish. This was accomplished by isolation of PGCs and profiling bulk cytosine DNA methylation levels using low coverage WGBS. This result will likely extend to other species that have a preformed germline including birds, amphibians and other fish. They also discovered amplification of the 45S rDNA, which coincided with loss of DNA methylation. Intriguingly, this locus was in a region that is known to be sex-linked.

Overall, this is an interesting and well-written study. The authors are mostly careful not to overstate their results, especially with regards to the amplification of the 45S rDNA. Currently, only an association between 45S rDNA and feminization is reported and there are clear exceptions numerous males were found to have similar molecular phenotypes. Line 19 in the abstract will need to be modified as it is not female specific if it occurs in males as well. Either way, this process is more prevalent in females than males.

We thank reviewer 1 for their positive analysis of our manuscript and for highlighting both our novel results along with our cautious interpretation of them.

We have altered the abstract to mention “oocyte-specific” amplification of fem-rDNA, which is technically more correct than “female-specific”. Further comments on this are given below.

I have a few comments that need addressing.

1. I can find no information on the number of samples, replicates or deposited data for this study?

The total number of samples analysed by high-throughput sequencing was 116. For the analysis of embryos, larvae and juvenile fish between 1-28 days, 3 replicates for EGFP -ve and +ve were obtained for each time point (1, 2, 4, 7, 11, 14, 21, 25, 28 dpf), except for EGFP +ve cells at day 28 when 5 replicates were analysed (n=56). To evaluate the gonad transformation period and sexual maturity (35, 40, 45, 50, 70 dpf), 3 replicates were obtained in a similar fashion for each sex (n=60). Additionally, vasa:EGFP +ve and -ve cells were sorted from 15 individuals between 35 and 40 days for quantitative PCR experiments (n = 3 EGFP -ve females and males and EGFP +ve males and n = 6 EGFP +ve 35 and 40 dpf females). We apologize for the non-inclusion of the repository number in the previous manuscript. The data is publicly available at GEO GSE122695. (<https://www.ncbi.nlm.nih.gov/geo/query/acc.cgi?acc=GSE122695>).

This information, in full, has been included in the revised Methods section.

2. Similarly, I can find no information on the bisulfite sequencing conversion rates, the number of sequenced reads and alignment scores. These are especially relevant for understanding if there is potentially a technical reason why the rDNA is being overamplified. The data in Figure 5 could be due to technical artifact given there is so

much biological variation in the results. There are males throughout all three sections, even with females there is a broad range of methylation/DNA amplification.

We thank the reviewer for pointing out this oversight - information regarding the number of sequenced reads, mapping efficiency and total unique mapped reads has now been added to supplementary table 1. As is common for bisulfite sequencing in vertebrate species, the frequency of non-CG methylation was used as a proxy for calculating the maximum non-conversion rate during bisulfite treatment. Using this metric, all libraries had a bisulfite conversion efficiency of at least 96.02%.

The reviewer raises the possibility that differences in fem-rDNA expansion observed in Figure 5 (now Figure 4) could be a technical artefact. While variation in timing of fem-rDNA expansion is expected during the 'juvenile phase' we would like to stress the point that amplification and demethylation were not found in any of the somatic controls, or any germline samples prior to gonad transformation. Given the consistency with regard to the amount and nature of input material throughout the experiment (i.e. 11-98 cells per fish, sorted into identical buffer), it is difficult to imagine how technical issues could account for the observed results.

The reviewer makes one further comment regarding males being found in all three fem-rDNA classes. To be clear, we only found one presumptive male within the amplified and demethylated class of fem-rDNA (ie. Group 3). It has been reported that males can commit temporarily toward femaleness during gonad transformation, and subsequently undergo degradation of perinucleolar oocytes to become definitively male (Wang et al. 2007). It is likely the amplification and demethylation observed in this single sample is a remnant of a collapsed feminization process, and should be considered an outlier. We have clarified this point in the discussion.

3. The emphasis on transgenerational inheritance throughout the entire manuscript is off-topic. There are no experiments executed to test transgenerational inheritance based on some environmental exposure. If the authors want to mention the possibility of TGI in the discussion this is fine, but there is no need to mention it in the abstract, intro, results, etc.

Following the reviewer's comment here we have removed any mention of transgenerational epigenetic inheritance (TGI) when we are summarising our results in the abstract and introduction, and do not mention it all in the results section itself. This will remove any doubt as to whether our experiments tested transgenerational inheritance following an exposure.

Nevertheless, we feel it is important to describe transgenerational epigenetic inheritance in the introduction so the reader understands the consequences of epigenetic reprogramming in mammals (i.e. it largely blocks TGI), and how this may relate to non-mammalian vertebrates which have a continuously defined 'preformed' germline.

4. Figure 1 seems better suited as a supplementary figure.

Considering this suggestion (and that of reviewer 2), we have decided to remove this figure from the manuscript.

5. Line 107, there is no data to support that the rDNA overamplification “helps orchestrate sex determination”. It’s purely a hypothesis at this point.

We completely agree with the reviewer’s observation here; while fem-rDNA amplification is associated with sex determination and is located in the only sex determining locus thus far discovered, more experiments are required to test causality. While the original sentence was not making definitive statements about causality “...(our results)..suggest rDNA helps orchestrate sex determination” we have nonetheless replaced it with a softer sentence “...suggests fem-rDNA amplification is implicated in sex determination”

6. Given the use of low coverage WGBS, the authors need to make clear in the discussion that there could be localized differences to methylation that are not detected by this method.

We have updated the MS to highlight the limitations of the low-coverage WGBS method used in our study to assess methylation at single copy-loci. Amongst these is the unambiguous statement in the results:

“Our low-coverage analysis cannot quantify methylation at single-copy loci...”

and for the discussion, recognition of how deep-coverage analysis could help discovery of differential methylation at single copy loci:

“While full-coverage sequencing may uncover sex-specific methylation at single-copy loci....”

Reviewer #2:

In the manuscript, “Zebrafish preserve germline epigenetic memory globally but demethylate and amplify sex-linked rDNA during feminisation”, Ortega-Recalde et al use low cell number bisulfite sequencing to assess methylation of zebrafish germ cells at a total of 14 stages of development between 1 and 70 days post fertilization. Using this data, they address a longstanding question in the field of chromatin biology/germ cell development, which is whether or not the methylation state of non-mammalian vertebrate germ cells is reprogrammed in the same way it is in mammals. They report that global methylation erasure was not detected at any stage assessed. Analysis of methylation at individual sequences across these time points was not provided with one exception. Very intriguingly, the authors find a region of zebrafish chromosome 4 that contains ribosomal genes and is associated with sex determination undergoes demethylation and amplification in females around 35 dpf.

Over all, the paper includes two important findings, that are likely to be of interest to many. However, both are analyzed at a fairly superficial level.

We would like to thank reviewer 2 for recognizing the importance of our two major findings, and how we have answered a longstanding question in the field of chromatin biology/germ cell development. They also highlight the likely broad interest of our work.

The reviewer also indicates that perhaps more could have been done to explore both of these results, and specifically mention that low-coverage sequencing does not allow analysis of single loci (except for fem-rDNA, which is amplified). We feel that 'superficial' is perhaps a harsh assessment - in order to be certain there was not global epigenetic erasure at any stage of germline development in zebrafish (and not restrict ourselves to developmental timepoints where erasure is seen in mammals), we felt it was important to comprehensively survey the earliest germline time point possible in our system (24 hpf) right through until sexual maturity. This involved in excess of 65 flow cytometry experiments, over 116 BS-seq libraries (technically challenging due to low cell number), many hundred fish and over two years to complete. While further sequencing may happen in future (especially for timepoints around 1-2 dpf, and also at sexual development) we feel this is beyond the scope of this study.

Nevertheless, to partially address this comment (and that of the comment below), we have performed an additional analysis of methylation at repetitive and non-repetitive sequences. While this provided some further interesting information (see more detail in the next rebuttal point), this did not uncover any global erasure of DNA methylation.

For the methylation analysis, while it is useful to know that global methylation erasure/reestablishment is not detected at the stages examined, data in figure 1 and table S1 seem to show relatively significant changes in methylation (for example methylation seems to be around 84% at 1 and 2 dpf but at 75% at 7 dpf). It would be useful to know if there are subsets of sequences that change their methylation status between these (or other) stages. The authors also don't comment clearly on how transposons/pericentromeric sequences were handled in their analysis. Were they excluded or included? If excluded, the caveats associated with doing so should be highlighted in the discussion.

As explained for reviewer 1, the low-coverage sequencing approach we used is not capable of quantifying methylation single loci and our initial analysis included the whole zebrafish genome irrespective of sequence type (both of which we now clarify in the revised manuscript). Nevertheless, as an example of a genomic subset, we have since added an analysis to test if transposons and other repetitive sequences change methylation relative to non-repetitive sequences.

We found that on average 54.31% (n=116, SD 1.01) of CG calls mapped to repeated regions (very similar to the overall repeat level of 52%, from Howe *et al.*, 2013). Not surprisingly, we found repetitive regions had greater methylation for all the samples compared to non-repetitive sequences (p <0.01, Wilcoxon signed-rank test), 86.94% (SD 1.88) and 67.64% (SD 4.29) respectively. Importantly, there was no hypomethylation of germlines samples in either repetitive or non-repetitive subsets relative to non-germline controls.

We have since updated the manuscript to reflect these new results and have been sure to explain where and when transposable elements were included in our analysis.

The second finding of sex specific demethylation and amplification on chromosome 4 is potentially very exciting, but exploration of this finding feels almost too limited to really appreciate its significance.

First, there is no secondary method of validation for what is a somewhat extraordinary observation. This is needed. Can the authors use Southern blotting to clearly demonstrate amplification of the repeat? (The repetitive nature of the region should counter any issues related to limited sample availability). Other PCR based methods could likely be used to confirm if Southern blots aren't technically feasible due to sample limitation.

While massive amplification of rDNA is an astonishing phenomenon, it is not completely unknown. The first reports date from 1968, where two groups (Brown and Dawid 1968; Gall 1968) found specific rDNA amplification in *Xenopus* oocytes. Subsequent studies in another species validated this finding (Motta, Andreuccetti, and Filosa 1991; Jabłońska et al. 2002; Thiry and Poncin 2005). The reason our finding is unique comes from the fact that expansion is so great specifically during female sexual determination.

Upon the suggestion of reviewer 2, we used quantitative PCR experiments to detect amplification of fem-rDNA in embryos 24 hpf, and 35- and 40-dpf. Using non-germline samples as a baseline for amplification, we found only females 35-40 dpf showed amplified fem-rDNA. Importantly, we found amplification was 2.73 - 93.54-fold greater than background – well within the range we found using low-coverage PBAT.

Second, while the link between amplification and demethylation and sex determination is tantalizing, the manuscript would be made stronger by further insights, such as additional data addressing mechanisms and /or consequences. A few potential questions that could be addressed are: Does exposure to the demethylating agent 5 azacytidine lead to amplification of this region in germ cells or in other contexts? Is demethylation/amplification triggered by Tet enzymes? Is demethylation and amplification accompanied by other chromatin changes? What does methylation of this region look like in wild zebrafish? Is demethylation of this region required for amplification/ transcription/sex determination. Exploring all these directions would likely be beyond the scope of a single manuscript. However, additional insights into some aspect of the biology associated with this finding would help to better place it in context.

We thank reviewer 2 for their detailed suggestions regarding future experimentation. In addition to the issue of scope mentioned by this reviewer, we would also like to highlight some conceptual/technical difficulties in addressing this issue.

Firstly, the reviewer suggests treating embryos with 5-azacytidine (5-aza) to examine if demethylation and amplification occurs. 5-aza is already known to drive feminisation of zebrafish, so one would expect higher fem-rDNA amplification in 5-aza treated embryos even if demethylated fem-rDNA was not directly responsible for feminisation. In our opinion, the only way to effectively address this question is to forcibly demethylate fem-rDNA, but not other sequences (unfortunately, 5-aza is global).

[redacted]

We are also actively testing other aspects of fem-rDNA function and mechanism in an attempt to triangulate its role in sex determination. For example, we are using zebrafish to test how fem-rDNA becomes circularised and amplified. We are also asking if demethylated and amplified fem-rDNA can it be found in other species during sex determination, and what happens to fem-rDNA in sex changing fish and those with seasonal breeding. While these endeavours are promising, unfortunately none are sufficiently developed for publication.

Minor:

In Fig 2 It is hard to see the germ cells perhaps a zoom in or higher magnification would be helpful

We have modified Figure 2 (now Figure 1) and Supplementary Figure 1 to improve the visualization.

Personally, I found figure 1 to be fairly confusing, I would suggest omitting.

As for reviewer 1, who also mentioned issues, we have removed Figure 1 from the manuscript.

Reviewer #3:

General comments

This ms addresses whether vertebrates with a “pre-formed” germline (most species) demethylate DNA during early development as it occurs in mammals and urodele amphibians, which have an “induced” germline and, according to studies in mice, exhibit two waves of global demethylation during early development. Here, using zebrafish they find that there seems to be no demethylation events in the germline.

One of the major problems of this study is that no methylation data is provided within the first 24 h post fertilization. If embryos were collected at 1.5 h post fertilization (hpf; Fig. 2A) why then CG Methylation was measured only starting at 24 hpf? In fact, during the first 24 h after fertilization important events take place and it is likely that by measuring CG methylation only starting at 24 hpf these events may have been missed. One can argue that there are previous studies covering the first 24 h in zebrafish, but this opportunity should be taken to repeat those analysis. Not providing robust data concerning the first 24 h can invalidate the main conclusion of this paper. Thus, samples should be taken also at fertilization and with the first 24 subsequent hours in addition to other time points.

Unfortunately, it is technically not possible for us to isolate germline cells prior to the first 24 hpf using the vasa:EGFP line. The reason for this is that background oocyte-derived vasa:EGFP is retained in the early embryo in all cells, despite no active transcription. A clear difference between germline and soma with respect to vasa:EGFP is only apparent at 24 h and afterwards.

Nevertheless, given that all other global demethylation systems so far studied take much longer than 1.5 days to demethylate and then remethylate (i.e. post-fertilisation embryo, primordial germ cells and the cultured naïve ES cells of mammals), we considered it highly unlikely global erasure would occur during this time and remain undetectable at 24 h. This result has now confirmed by Bogdanovic and colleagues, who sampled the *kop-EGFP-F-nos3'UTR* transgenic line PGCs between 4 and 36 hpf and found no global, or significant local, demethylation (Skovortsova, *et al*, 2019).

Specific comments

Between lines 119 and 122 there should be a description on the methods used to sacrifice the fish, the approved protocol, etc. All these details are missing.

We thank the reviewer for highlighting the missing method of euthanasia, and have now corrected it in the revised manuscript. The animal ethics approval was ET 25/2017 as listed previously.

Line 178. Why the better assembled version 11 of the zebrafish genome (GRCz11) was not used?

Initially, GRCz10 was considered for the analysis because the genome packages available at Bioconductor included just this and previous assemblies. To improve the manuscript, we created a new BSgenome data package with the latest genome assembly (GRCz11) and repeated all the analysis, including bootstrap sampling and analysis of previous datasets. We did not find any significant differences in our results. All the analyses, figures and supplementary data were updated accordingly.

Line 209. “mentioned above”, not “mention above”. Line 268. Use “PGC” instead of “primordial germ cells”. Line 342. “It has been recently reported that...” Add citation. Line 372. “Many germline samples”. How many? Were they males or females? Line 374. “... in some females...” How many? Why not all of them. End of line 387. Add “(0-30%)”. Line 413. “species with an induced germline” should be “species with a pre-formed germline”.

All of these typographical errors and missing data have been addressed in the revised manuscript as suggested.

Reviewer 3 also asks here why amplification does not occur in all female samples tested. The peak of fem-rDNA amplification appears to be subject to a high degree of variation in time, just as sexual commitment in zebrafish also appears to be variable in timing (Wang et al. 2007). While we expect that all females will undergo fem-rDNA expansion at some stage, it is impossible to know exactly when this will be for each individual fish.

Line 470. The data of this ms does not support the claim that “fem-rDNA” is further implicated in sex determination”. At best it shows some sort of correlation, which does not mean causation.

Like reviewer 2, we agree with this statement and have updated the manuscript accordingly to this observation.

Reviewer #4:

Ortega-Recalde and colleagues present an interesting study which characterises the DNA methylome during zebrafish germ line development. They do not identify a global DNA demethylation event, as documented in mouse and human – an important and provocative observation. In addition, they present evidence of rDNA amplification in female germ cells – and suggest this might be related to the previously uncharacterised mechanism of sex determination in zebrafish.

Overall this is an interesting and well-conducted study, by a group with a strong pedigree in the DNA methylation field. It will be of general interest, in particular to researchers studying germ line development and epigenetic reprogramming. However, there are some issues that need to be addressed prior to publication.

We thank the reviewer for identifying the importance of our work, and the robust nature of our experiments. It is very much appreciated.

1. rDNA amplification. The possible connection to sex specification is interesting. While it is not necessary that the authors fully uncover the mechanism, I think further discussion, or a model, of how this might work should be included. If rDNA is mechanistically linked to sex specification in germ cells then would this not lead to a progressive increase in rDNA over generations? How do the authors imagine this works?

While we could have been clearer about this in the manuscript, fem-rDNA in zebrafish is thought to be magnified ‘extrachromosomally’ by rolling circle amplification (Locati et al. 2017). Presumably replication of extrachromosomal fem-rDNA can be uncoupled from genomic DNA replication (by expressing or silencing rolling circle amplification machinery in isolation of regular DNA replication machinery). Because fem-rDNA replication could be shut down, it is unlikely to accumulate over subsequent generations.

We have since made clear that fem-rDNA is amplified extra-chromosomally, as in sentence beginning:

“Extrachromosomal amplification of rDNA by rolling circle intermediates....”

2. Epigenetic reprogramming vs DNA demethylation. Throughout the manuscript the authors tend to use these terms interchangeably (for instance ‘ Results: Line 267 ‘Absence of genome-wide epigenetic reprogramming in the zebrafish germline’. Repeated line 417). This paper deals specifically with DNA demethylation and so the authors should not broaden their claims. It is possible that there are global changes in other epigenetic marks (histone modifications or histone variants), perhaps in a manner akin to the mouse

germline, that occur independently of DNA demethylation in zebrafish. All instances should be corrected, and the authors may wish to discuss this issue.

We agree with the reviewer's comment here. We have revised all instances where we imprecisely used the term 'epigenetic reprogramming' to discuss DNA methylation reprogramming.

3. The quantification of DNA methylation is based solely on WGBS analysis. The merits of this as a methodology to assess global DNA methylation levels is disputed. In particular, the presence of repetitive elements with unknown copy number is problematic – and they likely represent a significant proportion of global DNA methylation. Have the authors validated their findings with an orthogonal quantitative method, such as LC-MS?

Performing DNA methylation analysis on low cell numbers remains technically challenging.

Other than PBAT, LC-MS is the only method we are aware of that can accurately quantify genomic methylation from low cell numbers (Amouroux et al. 2016; Wyck et al. 2018). Nevertheless, it is unclear if it would work on our samples (11-98 cells) – the only reports we could find operating near this scale used approximately 100 cells or more.

Apart from the obvious technical challenges (including having access to appropriate mass-spectrometers), the suitability of LC-MS in our experimental context is questionable. CG methylation is the only form of vertebrate methylation that has the capacity to transmit epigenetic memory in replicating cells (this is due to the palindromic nature of CG dinucleotides). In order to detect if epigenetic memory in the form of DNA methylation is erased, it is essential that methylation in the CG context is measured accurately. Unlike PBAT, LC-MS cannot distinguish between CG and non-CG methylation.

As reviewer 4 points out, one advantage of LC-MS is that it can quantify methylation from regions of the genome which have not yet been sequenced. Zebrafish represents a small and well-covered genome comprising 1.67 Gbp of assembled sequence (see <https://www.ncbi.nlm.nih.gov/grc/zebrafish/data>). Predictions of genome size using flow cytometry and bulk fluorometric assays are very close to this value (1.68-1.8 Gbp) (Ciudad et al. 2002; Vinogradov 1998; Wei et al. 2003; Hinegardner and Rosen 1972), meaning it is unlikely that a considerable portion of the genome remains unsequenced in zebrafish.

4. The authors present convincing data that there is no significant global DNA demethylation at the stages analysed. Is it possible that DNA demethylation occurs at a later or (perhaps less likely) an earlier stage? For instance,
 - given the low level of 5mC in eggs, presumably there must be DNA demethylation at some point during oogenesis? When does this occur?
 - Might it be that DNA demethylation occurs in sex specific manner in zebrafish? This may be different in zebrafish due to the apparently enigmatic mechanism of sex determination.
 - is it possible that there is also a period late DNA demethylation followed by DNA remethylation in the male germ line? Can the authors formally rule this out?

- alternatively, is it possible that there is early DNA demethylation and subsequent remethylation that occurs in the earliest specified PGCs?

Despite our efforts to comprehensively characterize global methylation throughout the germline, the very start and end of germline development were difficult to test for technical reasons. As mentioned previously in response to reviewer 3, *vasa:EGFP* presence in somatic cells before 24 hpf means isolation of cells during this period is impossible using our transgenic line. However, given temporal constraints in DNA demethylation and remethylation, we considered it highly unlikely that global methylation erasure occurred prior to 24 hpf. This result has been confirmed by Bogdanovic and colleagues using the *kop-EGFP-F-nos3'UTR* transgenic line (Skovortsova, *et al*, 2019).

On the other hand, we were unable to test both female and male germline cells in the last phases of gametogenesis. Oocytes greater than 40 microns were filtered out of our flow cytometry experiments to minimise the risk of blockages, and male cells in late spermatogenesis lose *vasa:EGFP* expression. Although we cannot formally rule out some demethylation and remethylation during these sex-specific phases of differentiation, we consider this is very unlikely to involve full erasure and remethylation that is not at least partially detectable in the samples we tested.

5. If there is no reprogramming of DNA methylation, is there evidence of more variability in DNA methylation patterns between zebrafish individuals (as compared to mice). Would the authors expect differences to emerge in different populations (or across generations), as any sporadic accumulation of DNA methylation would not be reset in the germ line? Might this impact the CG content in the genome over evolutionary time?

The question the reviewer raises is an interesting one – would we expect to find that methylation accumulates in a species without germline epigenetic erasure? To answer this, one could turn to activated cells in culture, which are ‘immortal’ like the germline and do not undergo periodic global erasure and remethylation. While some methylation does accumulate in cultured cells (Ziller *et al*. 2013), this never occurs to saturation as we might expect. Presumably this is because some level of active and passive demethylation partially counteract background *de novo* methylation in a balanced manner. It is likely that over time zebrafish would naturally find its own equilibrium defined by the interaction between this opposing methylation machinery.

6. Introduction:

Page 2

- PGCs are not stem cells. They do not self-renew
- the cells from which PGCs are induced are not somatic, they are pluripotent progenitors.

We thank the reviewer for picking up these errors. We have replaced ‘somatic’ with ‘epiblast’ to be correct about the progenitors of PGCs, and now do not refer to them as stem cells.

- Abstract and introduction: ‘no global barrier to transgenerational epigenetic inheritance’. This is not necessarily the case. There may be locus specific mechanisms to erase DNA methylation (i.e. some barrier, albeit not complete). Erasure of DNA methylation may occur at a different stage (see point 4). There may also be erasure/reprogramming of other marks. On a related point, in mammals there is little evidence that DNA methylation leads to transgenerational epigenetic inheritance in any meaningful way (even when it escapes reprogramming). Other mechanisms have been proposed (such as tsRNA) - so there may be barriers to such mechanisms in zebrafish PGCs.

In line with comments from other reviewers, we have dramatically reduced our claims around transgenerational epigenetic inheritance – now restricting this to the discussion. We also no longer describe zebrafish as having ‘no global barrier to transgenerational epigenetic inheritance’.

7. The section 412 – 419 is confused/confusing. - xenopus and zebrafish do not have an induced germline. - commitment is a strange term here. - prominent – possible maybe? - it is worth noting that although preformation is more common (likely because it encourages speciation), epigenesis is the conserved mechanism

The section has been modified according to the suggestions from the reviewer and the typographical errors have been corrected as below:

“In contrast to mammals, species with a preformed germline such as *Xenopus* and zebrafish do not require *de novo* formation of PGCs at each generation and instead use inherited cytoplasmic determinants to continuously define germline cells (Johnson et al. 2011). In line with the lack of cellular reprogramming required, our study shows that global DNA methylation erasure is not a feature of germline specification in zebrafish (Figure 5A).”

We thank the author for pointing out that it has been hypothesised that epigenesis is the ancestral mechanism of PGC specification. However, given we have only analysed one species in this manuscript (zebrafish), we have avoided discussing what might be the ancestral form of germline epigenetics (i.e. erased or not).

8. Page 21: line 411. reference. ‘loss of 5methyl-cytosine in the early embryo is tightly linked to acquisition of naïve pluripotency’. The primary papers should be cited here Habibi et al., Leitch et al., Ficz et al. – all 2013.

References were added in the updated version of the manuscript (Habibi et al. 2013; Leitch et al. 2013; Ficz et al. 2013).

References

Amouroux, Rachel, Buhe Nashun, Kenjiro Shirane, Shoma Nakagawa, Peter Ws Hill, Zelpha D’Souza, Manabu Nakayama, et al. 2016. “De Novo DNA Methylation Drives 5hmC Accumulation in Mouse Zygotes.” *Nature Cell Biology* 18 (2): 225–33. doi:10.1038/ncb3296.

- Brown, D. D., and I. B. Dawid. 1968. "Specific Gene Amplification in Oocytes. Oocyte Nuclei Contain Extrachromosomal Replicas of the Genes for Ribosomal RNA." *Science (New York, N.Y.)* 160 (3825): 272–80. doi:10.1126/science.160.3825.272.
- Ciudad, Juana, Elena Cid, Almudena Velasco, Juan M Lara, José Aijón, and Alberto Orfao. 2002. "Flow Cytometry Measurement of the DNA Contents of G0/G1 Diploid Cells from Three Different Teleost Fish Species." *Cytometry* 48 (1): 20–25. doi:10.1002/cyto.10100.
- Ficz, Gabriella, Timothy A. Hore, Fátima Santos, Heather J. Lee, Wendy Dean, Julia Arand, Felix Krueger, et al. 2013. "FGF Signaling Inhibition in ESCs Drives Rapid Genome-Wide Demethylation to the Epigenetic Ground State of Pluripotency." *Cell Stem Cell* 13 (3): 351–59. doi:10.1016/j.stem.2013.06.004.
- Gall, J G. 1968. "Differential Synthesis of the Genes for Ribosomal RNA during Amphibian Oögenesis." *Proceedings of the National Academy of Sciences of the United States of America* 60 (2): 553–60. <http://www.ncbi.nlm.nih.gov/pubmed/5248812>.
- Habibi, Ehsan, Arie B Brinkman, Julia Arand, Leonie I Kroeze, Hindrik H D Kerstens, Filomena Matarese, Konstantin Lepikhov, et al. 2013. "Whole-Genome Bisulfite Sequencing of Two Distinct Interconvertible DNA Methylomes of Mouse Embryonic Stem Cells." *Cell Stem Cell* 13 (3): 360–69. doi:10.1016/j.stem.2013.06.002.
- Hinegardner, Ralph, and Donn Eric Rosen. 1972. "Cellular DNA Content and the Evolution of Teleostean Fishes." *The American Naturalist* 106 (951): 621–44. doi:10.1086/282801.
- Jabłońska, A, T Szklarzewicz, W Jankowska, M Kukiełka, and S M Biliński. 2002. "rDNA Amplification in Previtellogenic and Vitellogenic Oocytes of Symphylans (Arthropoda, Myriapoda)." *Folia Histochemica et Cytobiologica* 40 (1): 43–46.
- Johnson, Andrew D., Emma Richardson, Rosemary F. Bachvarova, and Brian I. Crother. 2011. "Evolution of the Germ Line-Soma Relationship in Vertebrate Embryos." *Reproduction* 141 (3): 291–300. doi:10.1530/REP-10-0474.
- Leitch, Harry G, Kirsten R McEwen, Aleksandra Turp, Vesela Encheva, Tom Carroll, Nils Grabole, William Mansfield, et al. 2013. "Naive Pluripotency Is Associated with Global DNA Hypomethylation." *Nature Structural & Molecular Biology* 20 (3): 311–16. doi:10.1038/nsmb.2510.
- Locati, Mauro D., Johanna F.B. Pagano, Geneviève Girard, Wim A. Ensink, Marina Van Olst, Selina Van Leeuwen, Ulrike Nehrdich, et al. 2017. "Expression of Distinct Maternal and Somatic 5.8S, 18S, and 28S rRNA Types during Zebrafish Development." *Rna* 23 (8): 1188–99. doi:10.1261/rna.061515.117.
- Motta, C M, P Andreuccetti, and S Filosa. 1991. "Ribosomal Gene Amplification in Oocytes of the Lizard *Podarcis Sicula*." *Molecular Reproduction and Development* 29 (2): 95–102. doi:10.1002/mrd.1080290202.
- Thiry, Marc, and Pascal Poncin. 2005. "Morphological Changes of the Nucleolus during Oogenesis in Oviparous Teleost Fish, *Barbus Barbus* (L.)." *Journal of Structural Biology* 152 (1): 1–13. doi:10.1016/j.jsb.2005.07.006.
- Vinogradov, A E. 1998. "Genome Size and GC-Percent in Vertebrates as Determined by Flow Cytometry: The Triangular Relationship." *Cytometry* 31 (2): 100–109. <http://www.ncbi.nlm.nih.gov/pubmed/9482279>.
- Wang, X. G., R. Bartfai, I. Sleptsova-Freidrich, and L. Orban. 2007. "The Timing and Extent of 'juvenile Ovary' Phase Are Highly Variable during Zebrafish Testis Differentiation." *Journal of Fish Biology* 70 (sa): 33–44. doi:10.1111/j.1095-8649.2007.01363.x.
- Wei, Wen-Hui, Jing Zhang, Yi-Bing Zhang, Li Zhou, and Jian-Fang Gui. 2003. "Genetic

Heterogeneity and Ploidy Level Analysis among Different Gynogenetic Clones of the Polyploid Gibel Carp." *Cytometry. Part A : The Journal of the International Society for Analytical Cytology* 56 (1): 46–52. doi:10.1002/cyto.a.10077.

Wyck, Sarah, Carolina Herrera, Cristina E Requena, Lilli Bittner, Petra Hajkova, Heinrich Bollwein, and Raffaella Santoro. 2018. "Oxidative Stress in Sperm Affects the Epigenetic Reprogramming in Early Embryonic Development." *Epigenetics & Chromatin* 11 (1): 60. doi:10.1186/s13072-018-0224-y.

Ziller, Michael J, Hongcang Gu, Fabian Müller, Julie Donaghey, Linus T-Y Tsai, Oliver Kohlbacher, Philip L De Jager, et al. 2013. "Charting a Dynamic DNA Methylation Landscape of the Human Genome." *Nature* 500 (7463): 477–81. doi:10.1038/nature12433.

REVIEWERS' COMMENTS:

Reviewer #1 (Remarks to the Author):

I am satisfied with the responses from the authors. My last recommendation is to make as many of the supplementary tables/figures available as part of this manuscript instead of being linked to outside sources.

Reviewer #2 (Remarks to the Author):

All of this reviewers comments and concerns have been sufficiently addressed.

Reviewer #3 (Remarks to the Author):

The revised version has incorporated most changes asked by the reviewers but the major two points I raised have not been resolved. The first major issue was that no data was presented for the first 24 h after fertilization, a period where reprogramming could be located. In their response, authors claim that it is not possible to measure DNA methylation in the germ cells before 24 h but combination of disaggregation and cell sorting should allow that. The point is that, in my opinion, it is too risky to conclude that there is no reprogramming based on data that misses the first day of life in zebrafish. In fact, there is published literature (Potok et al., 2013 Cell; Jiang et al., 2013; Cell) that shows that zebrafish achieve a totipotent chromatin stage by reprogramming the maternal genome to that of the sperm at the time of zygotic genome activation. Thus, the affirmation that there is no reprogramming in the germ cell line is risky. In Fig. 1 there is a change from around 75% methylation to 82% methylation between day 7 and 11 and then back to 75% by day 14 while during the same period somatic cells stay flat at 75%. Further, in Fig. 2 G,H changes in germ cell DNA methylation appear also to fluctuate around 5-10%. Changes of around 5% is what reported Jiang et al 2013, so I do not see how it can be affirmed that there is no reprogramming. Certainly differences in mammals are bigger, but perhaps differences between mice and zebrafish are in the amount of DNA methylation changes needed to achieve reprogramming.

Authors also improved the discussion about fem-rDNA amplification and toned down its possible association with sex determination. However, as they concur, there is no mechanistic insights to prove causality.

Reviewer #4 (Remarks to the Author):

The authors' rebuttal was detailed, clear and well-argued. I have no remaining major concerns. However, the following minor points need some attention.

1. Abstract: 'does not undergo genome-wide epigenetic erasure during development.' – consider changing to 'does not undergo genome-wide DNA demethylation...'

2. 'Global DNA demethylation in mammals appears to be intrinsically associated with the reacquisition of developmental potency in the germline and, following that, lineage commitment' – the meaning of this statement is unclear. In what sense does DNA demethylation impact potency of germ cells? Indeed, in the mouse it could be argued that the pre-gonadal stage of PGC development is more associated with pluripotency – and that cells commit to germline after colonisation of the gonad, concurrently with genome-wide DNA demethylation.

3. 'These results suggest epigenetic erasure between generations is not prevalent in non-mammals,' This statement seems too broad given the examples provided. I would replace 'non-

mammals' with 'fish'.

4. 'One of these clusters contains the canonical expressed in all somatic cells.' – is a word missing here

5. ' we find consistent methylation levels in both the male and female zebrafish germline from the gonad transformation period until the point of sexual maturity and the final stages of male gametogenesis' – As the maximum level measured by the authors in the male germline is approximately 80% even in their latest sample (70dpf) (Figure 2H) but total methylation approaches 95% in sperm (Figure 5A) it isn't clear that they have measured up to the final stages of male gametogenesis(?), nor is it clear that DNA methylation is consistent at the final stages (i.e. it is clearly higher in sperm). Maybe this statement should be removed or altered.

Reviewer #1

I am satisfied with the responses from the authors. My last recommendation is to make as many of the supplementary tables/figures available as part of this manuscript instead of being linked to outside sources.

We thank reviewer 1 for their comments and helping us improve the manuscript. We have not put any supplementary tables or figures on repositories or outside sources. Where outside sources have been used, this is inline with editorial policy as far as we are aware – in particular, we have used NCBI SRA for deposition of sequencing data and GitHub for source code. We could perhaps deposit the source code as a supplementary data file, however, GitHub is more convenient for users as they can simply clone the repository and associated data, and the scripts will work immediately.

Reviewer #2

All of this reviewer's comments and concerns have been sufficiently addressed.

We thank reviewer 2 for their constructive comments.

Reviewer #3

The revised version has incorporated most changes asked by the reviewers but the major two points I raised have not been resolved.

We would also like to thank reviewer 3 for the positive comment and recognizing the improvements in our manuscript.

The first major issue was that no data was presented for the first 24 h after fertilization, a period where reprogramming could be located. In their response, authors claim that it is not possible to measure DNA methylation in the germ cells before 24 h but combination of disaggregation and cell sorting should allow that. The point is that, in my opinion, it is too risky to conclude that there is no reprogramming based on data that misses the first day of life in zebrafish. In fact, there is published literature (Potok et al., 2013 Cell; Jiang et al., 2013; Cell) that shows that zebrafish achieve a totipotent chromatin stage by reprogramming the maternal genome to that of the sperm at the time of zygotic genome activation. Thus, the affirmation that there is no reprogramming in the germ cell line is risky. In Fig. 1 there is a change from around 75% methylation to 82% methylation between day 7 and 11 and then back to 75% by day 14 while during the same period somatic cells stay flat at 75%. Further, in Fig. 2 G,H changes in germ cell DNA methylation appear also to fluctuate around 5-10%.

Changes of around 5% is what reported Jiang et al 2013, so I do not see how it can be affirmed that there is no reprogramming. Certainly differences in mammals are bigger, but perhaps differences between mice and zebrafish are in the amount of DNA methylation changes needed to achieve reprogramming.

In the revised manuscript, we have been very careful not to describe zebrafish as lacking reprogramming or DNA demethylation. Our data does not show this, and we would be misrepresenting to describe it otherwise. Nevertheless, we feel justified in claiming that zebrafish does not undergo complete genome-wide DNA methylation erasure like mammals. On this point, the view of reviewer 3 (and the other reviewers) is no different to our own.

To address the additional points Reviewer 3 makes here:

Technical limitations prevented us from isolating PGCs before 24 hpf. The reason for this is not related to the disaggregation or cell sorting protocol, but with the unspecific expression of the transgenic cell marker (*vasa:EGFP*) before 24 hpf. *vasa:EGFP* is highly expressed in oocytes and remnants persist in the early zebrafish embryo despite no active transcription. In addition to being highly unlikely due to temporal constraints, a co-submitted manuscript in this journal (Skovortsova, *et al*, 2019), used an alternative transgenic line to unequivocally show there is no global erasure of DNA methylation during the first 36 hpf of germline development. We have now clarified the inability of our model to probe this timepoint in the discussion, and have briefly summarised the other significant findings from their work.

Authors also improved the discussion about fem-rDNA amplification and toned down its possible association with sex determination. However, as they concur, there is no mechanistic insights to prove causality.

We are pleased the reviewer recognises here that we accurately describe the association of rDNA with sex determination (and not causality). Again, we do not feel the reviewers viewpoint differs from our own.

Reviewer #4

The authors' rebuttal was detailed, clear and well-argued. I have no remaining major concerns. However, the following minor points need some attention.

We thank the reviewer 4 for their detailed and insightful comments and helping us improve the manuscript.

1. Abstract: 'does not undergo genome-wide epigenetic erasure during development.' – consider changing to 'does not undergo genome-wide DNA demethylation...'

The manuscript has been updated according to reviewer 4 suggestion so that the distinction between epigenetic memory and DNA methylation is clear. However, we think it is a mistake to not include the word 'erasure' as this implies complete/near complete loss of DNA methylation, as opposed to DNA demethylation which is less explicit.

2. 'Global DNA demethylation in mammals appears to be intrinsically associated with the reacquisition of developmental potency in the germline and, following that, lineage commitment' – the meaning of this statement is unclear. In what sense does DNA demethylation impact potency of germ cells? Indeed, in the mouse it could be argued that the pre-gonadal stage of PGC development is more associated with pluripotency – and that cells commit to germline after colonisation of the gonad, concurrently with genome-wide DNA demethylation.

Hargan-Calvopina et al., (2016) showed that DNA methylation erasure in the germline of mammals is essential for preventing precocious differentiation of germline cells, and ultimately, safeguards germline potency. While we don't think our original statement was incorrect, we have changed it to better describe the Hargan-Calvopina result, and those summarised in the co-cited Surani review.

"Global DNA demethylation in the mammalian germline occurs in sexually undifferentiated PGCs, and is essential for safeguarding against precocious germline differentiation^{7,8}."

3. 'These results suggest epigenetic erasure between generations is not prevalent in non-mammals,' This statement seems too broad given the examples provided. I would replace 'non-mammals' with 'fish'.

We agree with reviewer 4 suggestion. We have incorporated the change in the revised version of the manuscript.

4. 'One of these clusters contains the canonical expressed in all somatic cells.' – is a word missing here

We thank reviewer 4 for spot a missing word in this sentence. It now states: "One of these clusters contains the canonical rDNA expressed in all somatic cells".

5. ' we find consistent methylation levels in both the male and female zebrafish germline from the gonad transformation period until the point of sexual maturity and

the final stages of male gametogenesis' – As the maximum level measured by the authors in the male germline is approximately 80% even in their latest sample (70dpf) (Figure 2H) but total methylation approaches 95% in sperm (Figure 5A) it isn't clear that they have measured up to the final stages of male gametogenesis(?), nor is it clear that DNA methylation is consistent at the final stages (i.e. it is clearly higher in sperm). Maybe this statement should be removed or altered.

We thank the reviewer for pointing out this inconsistency in our manuscript. We did not measure methylation levels at final stages of gametogenesis and methylation levels remain consistent except in mature sperm. Our comparison regards methylation levels in early stages of gametogenesis. We have modified the manuscript according to reflect this point:

“These gamete-specific methylation patterns appear to be generated relatively late in development compared to mice (Figure 5A) – we find consistent methylation levels in both the male and female zebrafish germline from the gonad transformation period until the point of sexual maturity for cells in early stages of gametogenesis.”